# Progress in Veterinary Behavior in North America: The Case of the American College of Veterinary Behaviorists

**DOI:** 10.3390/ani10030536

**Published:** 2020-03-24

**Authors:** Debra Horwitz, Katherine A. Houpt

**Affiliations:** 1Veterinary Behavior Consultations, 253 S. Graeser Rd., St. Louis, MO 63141, USA; debhdvm@gmail.com; 2Department of Clinical Sciences, Cornell University, Ithaca, NY 14853, USA

**Keywords:** clinical behavior problems, board certification, specialization

## Abstract

**Simple Summary:**

The American College of Veterinary Behaviorists is a specialty group within the American Veterinary Medical Association. It was formed by eight veterinarians and has grown ten-fold in the following decades. The specialty ensures that those who are its diplomates have taken the training, seen hundreds of cases, published research on animal behavior, and successfully passed an examination so the public can be assured that their animal will get the best treatment for its behavior problem.

**Abstract:**

The American College of Veterinary Behavior has grown in number and in expertise over the past quarter century. There are now 86 diplomates, at least three textbooks on treating behavior problems, and a text on veterinary psychopharmacology. Although veterinary behavior began in veterinary colleges, the majority of residents are now trained in non-conforming programs. Many more diplomates practice privately in specialty clinics or as separate businesses. Progress has been made in both diagnosis and treatment with polypharmacy, resulting in successful outcomes for many dogs and cats suffering from separation anxiety, fear, or aggression.

## 1. Introduction

While the study of basic animal behavior had been around for decades, the study of pet animals and the behaviors they exhibit that their owners do not want was a relatively unexplored discipline. The discussion of Veterinary Behavioral Medicine as a discipline to add to veterinary medicine began in the early 1980s. In the US there were several veterinarians employed at veterinary schools teaching animal behavior. They were at Texas A&M, University of California at Davis, University of Pennsylvania, Georgia, and Cornell. By then, cats and dogs had become “family members” and shared the home with the family. What followed were behavior issues that needed resolving. What the students wanted to learn was not the neurohumoral basis of feeding behavior, for example, but what kinds of behavior problems they would see in practice. We began to see cases as soon as the clinicians realized that there was someone who could help dog and cat owners with their pets’ behavior problems. The problems were unruly behaviors, aggression toward family and unfamiliar people, house soiling in dogs and mainly house soiling in cats.

Because this was a new field, we wanted to get together to share experiences and learn from each other, so we began to meet annually. One of the big decisions was whether to include non-veterinarians in the group, many of whom were seeing (and curing) behavior problems. The American Society of Veterinary Ethology (ASVE) was founded in 1976 and in 1986 was renamed the American Veterinary Society of Animal Behavior (AVSAB), and is still in existence today. Any veterinarian can join and others with advanced degrees can become adjunct members. In Europe, The Society for Veterinary Ethology was created in Edinburgh on 4 June 1966. It rapidly expanded to cover all applied aspects of ethology and other behavioral sciences relevant to human-animal interactions, such as farming, wildlife management, the keeping of companion and laboratory animals, and the control of pests. The Society also quickly became increasingly international: it now has an international federal structure as well as regional representatives around the world. In 1991, on the 25th anniversary of the SVE, the society was re-named the International Society for Applied Ethology (ISAE).

AVSAB was an interest group and forming an interest group is easy. Find a number of people with a common interest, write a constitution, hold meetings, charge dues, and you are a Society. AVSAB continues to this day, but before long veterinary behaviorists knew that a more formal and disciplined structure was needed, as we grew both in knowledge and size.

## 2. Growing a New Veterinary Specialty College

The process of becoming a veterinary specialty takes determination and a devoted group of likeminded individuals. In the United States within veterinary medicine, the American Veterinary Medical Association controls the creation and recognition of specialty colleges. To begin, you must gather a small nucleus, the founders or charter members, create a document that describes what you are and what you will do, and then you have to convince the American Board of Veterinary Specialists that Behavior is a legitimate specialty. The American Board of Veterinary Specialists consists of about a dozen veterinarians from other already approved specialties such as surgery and internal medicine. We had to provide the following information: Number of potential candidates in the near future;Plans for existing training programs, both conventional (university residencies) and non-conforming (private practice residencies), and what a resident will need to do in each;Plans for incorporation as 501 c organizations;Credentialing and examination process and plans.

The American College of Veterinary Behaviorists was approved as an official specialty of the American Veterinary Medical Association in 1993. The European College of Animal Welfare and Behavioural Medicine received provisional recognition from the European Board of Veterinary Specialisation (EBVS) in 2002 and full recognition in 2012

There were only eight people on the founding committee: Bonnie Beaver from Texas A&M, Katherine Houpt from Cornell, Benjamin Hart from University of California Davis, Victoria Voith from the University of Pennsylvania, Sharon Crowell-Davis from the University of Georgia, and several veterinarians chosen for their expertise in fields other than behavior. Tom Wolfle was a laboratory animal specialist, board-certified by the American College of Laboratory Animal Medicine, Elizabeth Schull was board-certified in Neurology, and R.K. Anderson was board-certified in Public Health.

As a new college we had many tasks and only eight people to do them. We had already determined that in order to be board-certified a veterinarian must have at least one year of clinical medical experience after veterinary school and then a two- to three-year residency, during which the major portion of time is devoted to treating animals with behavior problems, augmented with course work in various aspects of behavior, such as ethology, learning, and neuroscience. In addition to the residency, the veterinarian must publish a paper in a refereed journal, submit three acceptable case reports and take a two-day written examination. See Pankratz et al. (2017) and Albright et al. (2015) for examples of scientific papers that meet the criteria [1,2].

## 3. The Early Years of Board Certification

Initially, we had three residency programs, each taking one individual. Training was usually three years or more if a MS and/or PhD were also pursued. We were the first board certifying organization for behavior, and initially the growth of the college was slow. We always had a “non-conforming” track for those whose training pre-dated the formation of the college and those not in a university residency program. Over time we have refined and expanded our non-conforming training programs to meet the growing desire and need to have more veterinary behaviorists. At the present time most of the residents are in training programs not situated within the university setting. This has allowed our college to grow from 16 in 1996 to 86 today. The growth of non-traditional programs has allowed the American College of Veterinary Behaviorists to expand and broaden our reach across North America and even into foreign countries.

## 4. Changing Terminology to Fit the Pet Owners’ Situation

Initially, diagnostic terminology utilized broad terms and even borrowed some terms utilized in other animal behavior disciplines based on observations of many species, including chickens, wolves, and horses. The issues that arise in canine behavioral consultations are generally related to the interactions between the caregiver and the pet; therefore, the early focus was an attempt to understand and reorganize the human-canine relationship. Dogs communicate with each other using a system of assertive and subordinate postures to establish who prevails in a social conflict, generally over resources. In animal behavior literature, the term “dominance hierarchy” is used to denote a social ranking within a group in which some individuals give way to others in a social encounter or access to scarce resources [3]. This led, in the early days, to a diagnosis utilizing the term “dominance”, especially in regards to aggression by the dog toward familiar people. While dogs do communicate with other dogs in this manner, this is a poor model for explaining canine aggression toward familiar people, as we realized that in most cases the dog was acting like a dog, but aggression was driven by anxiety when the normal canine signals were misunderstood by the humans involved. This change in understanding was also borne out by the inappropriate use of physical dominance by popular, but ill-informed trainers, to sanction punishment-based training. This led veterinary behaviorists worldwide to use a more accurate terminology that was not based on dominance theory for human-directed aggression. Between 1991 and 2001, 667 of 1644 dogs were diagnosed with dominance aggression at Cornell University Behavior Clinic, whereas at the same clinic only 2 of 1931 dogs have been diagnosed in the years since then [4].

The dog is not trying to dominate, but is trying to communicate using the only language they know—facial expressions, body postures, and vocalizations. Because anxiety and fearful postures were recognized in the situation, many terms have evolved, but none are standard across behavioral organizations. Some terms utilized recently include “conflict aggression” [5], a descriptive diagnosis “aggression directed toward familiar people”, and finally, “social conflict aggression”. What we can all agree upon is that aggression toward familiar people is usually about the social interaction between the dog and the human and is fraught with misunderstandings of the other’s meaning and intent. What is also clear is that the dogs are not trying to “dominate” us, but are trying to decrease the conflict involved in dog-human interactions [6].

Another change in diagnosis was the realization that many dogs and cats are living in environments that do not meet their needs, resulting in anxiety and frustration. When veterinarians began to see behavior cases many dogs and cats were allowed free access to the outdoors, allowing them to engage in many of their species appropriate behaviors and meet many of their mental and physical needs. Often the dogs and cats were not alone all day. However, over the last 10–15 years that has changed. Most dogs are alone at home all day, perhaps without another dog to interact with. Many cats are no longer allowed outside, depriving them of hunting and play activities that are vital to their welfare. This has led to an increase in diagnoses related to anxiety (e.g., separation anxiety in dogs, anxiety-based aggression problems in cats), and frustration with the inability to engage in normal species behaviors and forced social interactions within their living situations, which the pet did not choose.

Another change in veterinary behavioral medicine within the USA is the realization that many pets relinquished to shelters and rescue situations could find homes with proper placement and training. However, due to vigorous spay/neuter program, a lack of dogs in one area of the country compared to another has led to dogs sourced from other places. Often, they have not received proper socialization/training or veterinary care and are placed in homes with inexperienced pet owners without the needed support to create a strong and lasting bond. Ongoing research is taking place to optimize placement of these pets and increase their chances of finding life-long homes. Improving the welfare of shelter dogs has long been the goal of diplomates of the American College of Veterinary Behaviorists [7,8].

## 5. Veterinary Behavior Technicians

Treatment of almost all behavior problems in dogs requires behavior modification with or without the addition of psychoactive medication. It is very helpful to have an assistant who can help the owner with behavior modification. Dog trainers can be hired, but the formation of the veterinary technician specialty in behavior, the Academy of Veterinary Behavior Technicians, has led to a pool of individuals who are not only experts in handling dogs, but have also demonstrated a thorough knowledge of learning principles. Use of those skilled in positive reinforcement and other non-aversive techniques allow the veterinary behaviorist time to determine the most likely diagnosis and most promising medication for the patient, while the technicians help the owner with behavior modification.

## 6. Changes in the Use of Psychoactive Medication

In the beginning owners were very reluctant to give psychoactive medication to dogs or cats. Their attitude was “just say no to drugs”. There seemed to be two objections to the use of medication: (1) the owners did not want to admit to others that they were using drugs to control their dogs; (2) they did not want to change the dog’s personality (even if that personality was vicious). Over the years, owners have become more educated about anxiety-related behaviors and are more likely to demand drugs than to refuse them. The other change was the development of three drugs with FDA approval for treating canine behavior problems. The efficacy of each of these drugs was proven by the research sponsored by the pharmaceutical companies and performed mostly by ACVB diplomates. The first was deprenyl (Anipryl^®^_,_ Parsippany, NJ, USA) selegiline), for the treatment of cognitive dysfunction in dogs [9].

The second was clomipramine (Clomicalm^®^Greensboro, NC, USA), for separation anxiety in dogs [10]. The third was fluoxetine (Reconcile^®^ Pensacola, FL, USA), for separation anxiety in dogs [11]. So far, no pharmaceutical company has offered a drug for aggression, although that is the problem most frequently presented to veterinary behaviorists. Apparently, the liability to the company if a dog bites while taking the drug is too great.

The number of studies done on the efficacy of these brand name drugs or their generic equivalent is small. One example is a study in which buspirone, a serotonin_1A_ partial agonist, reduced spraying in half the cats treated [12]. Dodman et al. (2000) used fluoxetine to decrease what was then called dominance aggression (aggression to owners) [13]. Pryor et al. (2001) showed that fluoxetine was very effective in reducing spraying by cats [14]. Recently, Chutter et al. (2019) showed that 65% of the dogs improved (were less aggressive) when they were treated with fluoxetine [15].

Most of the medications used for behavioral problems are given orally, but dexmedetomidine (Sileo^®^ Turku, Finland) is available as a gel to be given transmucosally. That route of administration leads to a very rapid onset of action. For that reason, it is very useful for treating storm and other noise phobias in dogs.

The next big step in treatment was combining drugs. Gruen et al. (2007) found that adding trazodone improved the behavior of dogs suffering from separation anxiety who were also receiving clomipramine [16]. Ogata and Dodman found that adding the alpha_2_ agonist clonidine, in addition to an Specific Serotonin Reuptake Inhibitor (SSRI) or tricyclic antidepressant, reduced fear-based behavior problems [17].

Now, it is not uncommon to use three drugs on our referred cases to reduce anxiety [18]. Trazodone is used for its immediate effect, while the SSRI prescribed has not yet exerted its full effect. A benzodiazepine or an alpha adrenergic blocker may be added. If there is any indication that pain is involved, gabapentin is prescribed.

The veterinary psychopharmacology text by Crowell-Davis et al. (2019) has added, in the second addition, chapters on miscellaneous serotonergic agents, anti-convulsants and mood stabilizers, N-methyl-D-aspartamate receptor agonists, and drug combinations [19].

## 7. In Conclusion

Over the past thirty years the American College of Veterinary Behaviorists has been able to improve the outcomes of behavior problems through better diagnostic procedures and, in particular, more effective use of psychoactive medication. Our progress is reflected in the textbooks on clinical animal behavior published by diplomates of the American College of Veterinary Behavior [20,21,22].

## 8. The Future

We look forward to specialty groups in behavior forming in other regions of the world. Below are some guidelines for those establishing certification for veterinarians specializing in treatment of behavior problems of animals.

(1) Establish flexible bylaw rules for establishing training programs that can be altered by the voting members of the specialty group;

(2) Account for people whose training predates the formation of the specialty organization;

How many will be grandfathered (exempt from training provisions) and why?

How long will certain options exist for providing alternative training paradigms once formal establishment of the specialty occurs?

(3) Make sure from the beginning to have alternate pathways to the same end, including university-based training programs and programs that take place within a private practice setting.

From the very beginning these programs should have the same requirements and oversight.

Update those requirements from time to time for both groups as the field changes;

(4) Keep good records and document how the training and testing have evolved to avoid confusion as the specialty group grows.

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
