# Peer review of "Progress in Veterinary Behavior in North America: The Case of the American College of Veterinary Behaviorists"

_animals, 2020, doi:10.3390/ani10030536_

Round 1

Reviewer 1 Report

The manuscript provides a historical process of veterinary behavior specialty development that would be useful information to be shared among veterinary medicine domestically and internationally.

I have found a few issues especially under the section of “Changes in Use of Psychoactive medication”. Once they are addressed that will improve the manuscript.

L17, “non-confirming”

The label should be consistent between non-conforming (L17) and non-traditional (L90).

L30, ---"not what the hypothalamus has to do with feeding behavior,”

It might be clearer to add what example was implied by this sentence. Was it an example of a traditional behavior lecture in physiology or something?

L33, “unruly behaviors”

It might be clearer to add examples of unruly behavior or to use alternately from “unruly”.

L41, the first word should be small letters

L41, (SVE) should be added after the official name.

L64, recommend removing “a notoriously difficult test” from this sentence.

L113----“to sanction punishment based training.”

It should be clarified either “positive punishment based training” or “physical punishment based training”

L147, ----“handling dogs,”

It would be better to use generalized population, such as “handling veterinary patients” or “handling pets”

L155 ----“to control their dogs.” Can be “to control their pets”

L156---“to change the dog’s personality” can be “to change the pet’s personality”

L156 “vicious” can be described alternatively such as “difficult to live with” or something

L157 “owners have become more educated and”

I would add clarification. For example, “owners have become more educated about anxiety related behaviors”

L159, I wonder if the authors intentionally did not include “Sileo®” as it was launched recently.

But can it be still included somewhere in this section?

L161-162, Please remove the space between the sentences.

L163, ---“for separation anxiety”

Please specify the species as it did so for Clomicalm.

L167, It seems to me to be a bit abrupt to start “Buspirone” following the previous sentence.

L167-L174. Each reference, such as Dodman et al. did not have publication year in the sentence.

Please follow the journal instruction.

L170, “the dogs improved---”

Please add what type of behavior problems in dogs were improved.

L171, This paper, Gruen et al. (2007), did not only report the cases of separation anxiety treated by clomipramine. Please revise this sentence matches to the cited reference.

L171, ---“Gruen et al. (2007) found”

L 173. “Ogata and Dodman found”

Since these studies were retrospective case-series and no control group were used, I would use “reported” instead of “found”.

L173, “alpha agonist” should be “alpha 2 agonist”.

L175-L178, “Trazodone---- is prescribed.”

Adding references to these three sentences are recommended.

L177, “adrenergic blocker”

This needs to be revised. Were you talking about alpha 2 agonist, or beta blocker, or both?

L 179, Can you add some connection between this sentence and the previous paragraph?

Or, can the L175-L181 be all revised together?

Author Response

Here is our response to the reviewer. The first number is the line number in the submitted manuscript and the second is the line number in the revised manuscript

L 17  L 21 non-confirming changed to non-conforming throughout ms.

L30 L36 changed ‘not what the hypothalamus has to with feeding’ to  ‘the neurohumoral basis of feeding’

 L 157 206 added about anxiety related behaviors

L 159 a228- 231added sentences about Sileo

  163 215 added species

167 220220 added a phrase to ease transition

L 167 -174  222=226 added years

L170- 226 added 'were less aggressive'

Reviewer 2 Report

The manuscript is a description of  the American College of Veterinary Behaviorists. It is not a scientific paper.

A number of statements are made about the development of the field (e.g. lines 107 to 110 on the used of dominance as an explanation for dog behaviour). While the development of the field may be interesting the description of the development needs to be funded on data, e.g. number of studies/clients/veterinarians using dominance as an explanation in the 80's compared with today.

Author Response

Added data from Cornell . Between 1991 and 2001 405 of 1191 dogs were diagnosed with dominance aggression  at Cornell University behavior clinic (Bamberger and Houpt 2006) whereas  only 2 of 1931  have been diagnosed in the years since then.

Reviewer 3 Report

I was prepared to give this paper a quick read and love it. This paper is listed as a REVIEW.  It is not.  See my comments on the ms. And to read some of them you may have to click on the highlights.  By the time the authors get to using any citations they do a woefully inadequate job of reviewing the evolution of thought and the field, including missing a lot of the literature contributing to that evolution. 

So my question at the beginning of this paper was the same as my question at the end....what was the point.

Author Response

  L 16 and 19 It is a review

29 added reviewer suggestion. Those with behavior training realized there was suffering because of problematic behaviors, clinicians, practitioners, and students wanted to do more, and emphasis shifted.

30 added Once help was apparent, they availed themselves of it.

41 moved paragraph

48 deleted paragraph

53 deleted sentence

56 added requirements for recognition by American Board of Veterinary Specialties and removed l 60- 70 and replaced with the following:

  1. Convincing evidence that the topic is a unique, active, and needed area of veterinary medicine (this is always the hardest part usually because of preconceived ideas)
  2. Constitution/Bylaws
  3. Number of potential candidates in the near future
  4. Plans for existing training programs both conventional (university residencies) and non-conforming (private practice residencies) and what a resident will need to do in each
  5. Plans for incorporation as 501 c organizations
  6. Credentialing and examination process and plans

80 re-wrote to eliminate first person plural

99 There are citations in the paragraphs

Reviewer 4 Report

The review is an interesting personal account of a veterinary field of specialisation. As such, there are areas that are not referenced as appropriate references are not available. 

Line 135- this paragraph on shelter and rescue organisation work does need some referencing as there are high quality studies available on behaviour and welfare in shelters.

The review is very US- biased and apart from reference to ISAE the information is exclusively from the US. There is also a European specialist behaviour group -  European College of Animal Welfare and Behavioural Medicine- and it would be good to refer to them as well in the article.

The diplomates publish a scientific article as part of their residency and it would be helpful to include a paragraph with reference to papers published in the last 2-3 years as examples of how the candidates are progressing the field of behavioural medicine. 

Line 9- capital letter for association?

Author Response

Added reference to European College of Animal Welfare and Behavioural Medicine , although our charge was to address only the American college

 Added two references as examples of publications in fulfillment of the College’s requirement for publication.

Round 2

Reviewer 2 Report

Although more information has been added and the paper has been improved it is still much more suitable for a journal interested in veterinary history than the current journal. 

Alternatively I would suggest that you submit it to a journal with a focus on the US. 

Author Response

This was an invited paper for a special issue of Animals so submitting it elsewhere is not an option.